# The Postpartum Obsessive-Compulsive Scale: Psychometric, Operative and Epidemiologic Study in a Portuguese Sample

**DOI:** 10.3390/ijerph191710624

**Published:** 2022-08-25

**Authors:** Ana Telma Pereira, Ana Araújo, Julieta Azevedo, Cristiana C. Marques, Maria João Soares, Carolina Cabaços, Mariana Marques, Daniela Pereira, Michele Pato, António Macedo

**Affiliations:** 1Institute of Psychological Medicine, Faculty of Medicine, University of Coimbra, Rua Larga, 3004-504 Coimbra, Portugal; 2Coimbra Institute for Biomedical Imaging and Translational Research, University of Coimbra, Azinhaga de Santa Comba, 3000-548 Coimbra, Portugal; 3Department of Psychiatry, Centro Hospitalar e Universitário de Coimbra, Praceta Professor Mota Pinto, 3004-561 Coimbra, Portugal; 4Center for Research in Neuropsychology and Cognitive-Behavioral Intervention (CINEICC), Faculty of Psychology and Educational Sciences, University of Coimbra, Rua do Colégio Novo, s/n, 3000-115 Coimbra, Portugal; 5Keck School of Medicine of the University of Southern California, 1975 Zonal Ave., Los Angeles, CA 90033, USA

**Keywords:** obsessive–compulsive disorder, postpartum, psychometrics, perinatal, obsessions, compulsions

## Abstract

Background: Although obsessive-compulsive (OC) symptoms are common in the perinatal period, measures to comprehensively assess their presence, frequency, interference and severity are lacking. The Perinatal Obsessive–Compulsive Scale (POCS) is the only self-report questionnaire with context-specific items. It includes items to assess perinatal-specific obsessions and compulsions, a severity scale and an interference scale. Objectives: (1) to analyze the validity and reliability of the Portuguese version of the POCS; (2) to find Obsessive–Compulsive Disorder (OCD) prevalence in postpartum and determine the POCS cut-off scores and its accuracy (sensitivity, specificity and predictive values) in screening for OCD according to DSM-5 criteria; (3) to describe the prevalence, content, severity, interference and onset of OC symptoms in the postpartum. Methods: 212 women in postpartum filled in a booklet, including the POCS Portuguese preliminary version, the Perinatal Anxiety Screening Scale and the Postpartum Depression Screening Scale; they were interviewed with the Diagnostic Interview for Psychological Distress—Postpartum. Results: Confirmatory Factor Analysis revealed that POCS presented acceptable fit indexes (χ^2^/df = 2.2971; CFI= 0.9319; GFI = 0.8574; TLI = 0.9127; RMSEA = 0.860, *p* < 0.001). The Cronbach’s alphas were all > 0.800. The POCS cut-off point that maximized the Youden Index (J = 0.86, 95% CI [0.94–0.99]) was 20, corresponding to an Area Under the Curve of 0.970 (*p* < 0.001; Standard Error = 0.031; 95% CI: 0.937 to 0.988). The prevalence of postpartum OCD was 3.30%. The severity of thoughts and behaviors was moderate to severe for approximately 15% of women. For thirty-five percent of women, the onset of symptoms was in the first three months postpartum. Conclusions: The Portuguese version of POCS has good validity, reliability and accuracy and may be considered ready for use in both clinic and research fields. POCS provides specific information regarding symptoms and individual patterns experienced by each woman, which allows normalization, destigmatization and personalized intervention.

## 1. Introduction

The risk for the development or exacerbation of Obsessive–Compulsive Disorder (OCD) is higher in the perinatal period, especially in the postpartum [1,2]. In addition to changes in the neuroendocrine milieu [3], the perinatal period lowers the threshold for Obsessive–Compulsive (OC) symptoms by presenting a sudden and intense increase in responsibility associated with motherhood [4].

Although one of the most common times for a woman to experience OCD and OC symptoms is after the birth of a child [5,6], postpartum obsessive–compulsive disorder (ppOCD), generally defined as first-onset or recurrence of OCD in the postpartum period, has been much less investigated than depression and anxiety in the same period. However, ppOCD has adverse effects on both mother and child and interferes in the mother-infant interactions and bonding [7].

OCD prevalence in the postpartum, approximately 3.0–7.0% [1,2,8,9], is higher than in the general population (1.08%) [5]. Even when the criteria for a full OCD diagnosis are not met, OC symptoms are more common in the perinatal period than in other times of a woman’s life [5,10]. In the postpartum context, unwanted, intrusive thoughts of infant-related harm are reported by nearly all new mothers (between 70 and 100%), with half of all new mothers reporting unwanted, intrusive thoughts of harming their infant on purpose [11]. The presence of such slightly intrusive and repetitive thoughts does not necessarily represent an obsession or lead to OCD [4]. Differences from OCD obsessions are their shorter duration, less distress, less difficult to dismiss and low association with compulsive behaviors [10].

In the perinatal period, the cognitions’ content tends to focus on the baby. The most common content of the repetitive thoughts involves accidents/aggression, such as the baby dying in her/his sleep, being harmed (e.g., choking, dropping, or inappropriate sexual contact with the baby) and contamination [12]. A recent meta-analysis showed that aggressive obsessions were much more common in ppOCD than in pregnancy-OCD and non-perinatal OCD [13]. Washing/cleaning compulsions were less frequent in ppOCD-PP, while infant-focused checking compulsions, self-reassurance and seeking reassurance from others were also more common in ppOCD [13].

Postpartum OCD is characterized by its rapid onset, a greater tendency towards aggressive obsessions, avoidance of obsessional cues (including avoidance of the newborn and knives), covert behaviors (thought suppression attempts and prayers) performed to neutralize the obsession and its association with depressive symptoms [4,12,14].

Although OC symptoms are common in the perinatal period, measures to comprehensively assess the presence, frequency, interference and severity of such symptoms are lacking, and further studies are needed on the use of screening instruments designed specifically for use in women with postpartum OCD [15]. Unlike some other disorders (such as major depressive disorder), DSM-5 [16] does not allow OCD to be characterized with the peripartum onset specifier. The lack of a specific subtype of OCD for the postpartum period in diagnostic classifications may have contributed to the deficient recognition of this clinical condition.

Until the development of the Perinatal Obsessive–Compulsive Scale (POCS) [17], the study of OCD in this period was hampered by the absence of a specific instrument. Because of the egodystonic nature of these infant-related intrusive thoughts, feelings of guilt and shame and the associated stigma, it is very rare for women to spontaneously report these symptoms, which are underdetected and undertreated [18], and may predispose to the development or exacerbation of OCD in vulnerable individuals [11]. These factors have led to recent calls for further screening of ppOCD in the postpartum period and the development of measures for this purpose [19].

The Yale-Brown Obsessive Compulsive Scale (Y-BOCS), along with the symptom checklist [20], is the most widely interview-based measure, both in clinical and research settings, to rate the severity of OC symptoms after identifying it with the Y-BOCS symptoms checklist. Despite the Y-BOCS’ strong psychometric properties, high internal consistency, good inter-rater reliability and construct validity [20,21,22,23], its checklist lacks context and content specificity for the perinatal period [17,24,25,26]. There is also a self-report paper-and-pencil version of the Y-BOCS, very similar to the rater administered version, with a list of symptoms and a severity scale for both obsessions and compulsions. It has demonstrated good reliability and validity but suffers from the same issue of not including specific perinatal worries [27]. Recently, Thiséus and collaborators (2019) adapted the Parental Thoughts and Behaviors Checklist (PTBC) [28] into a self-reported format for potential use as a screen for ppOCD, and its preliminary study found good psychometric properties. However, the PTBC items focus only on the cognitive dimension of ppOCD, lacking a comprehensive evaluation of the behavioral dimension/compulsions of OCD.

The POCS [17] was developed to apply specifically to the perinatal period; it has two versions: for pregnant and postpartum women. It includes items to assess perinatal-specific obsessions and compulsions, as well as a severity scale and an interference scale. The measure has been preliminarily validated in a mixed group of 162 pregnant and postpartum women, showing good psychometric properties. The POCS severity and interference scales’ Cronbach’s alphas were > 0.90, showing very good reliability. The correlation between the POCS and the Y-BOCS severity scales was significant and of high magnitude (*r* = 0.81, *p* < 0.001), which proves its concurrent validity. POCS’ sensitivity and specificity were >60% and >90%, respectively, with areas under the ROC (receiver operating characteristics) curve around 0.80, revealing its accuracy. However, the authors alerted that the clinical threshold should be interpreted with caution because it is not a validated cut-off score [17]. POCS also includes questions about the symptom’s onset and changes (better or worse) since postpartum. 

The (European) Portuguese version of the Prenatal Obsessive–Compulsive Scale has recently proved to be a valid and reliable instrument [29]. Using exploratory and confirmatory factor analyses with different samples of pregnant women, we have found that a measurement model composed of two dimensions—Severity and Interference—presented good construct validity. Internal consistency and convergent validity with perinatal anxiety and depression measures were also high. Nearly 10% of the women presented OC symptoms with relevant severity. Co-occurrence of obsessions and compulsions (44.4%) was associated with significantly higher PreOCS scores. Obsessions with the highest prevalence were about contamination, harm to the baby, others’ judgment and baby’s health; these increased the odds of having at least one compulsion. Accordingly, the most frequently reported compulsions were repeatedly searching for information, asking for reassurance, checking and cleaning.

Our aims were to (1) analyze the validity (construct, concurrent and discriminant) and reliability of the (European) Portuguese version of the Postpartum Obsessive–Compulsive Scale/POCS; (2) find the prevalence of OCD in postpartum and determine the POCS cut-off scores and its accuracy (sensitivity, specificity and predictive values) in screening for OCD according to DSM-5; (3) describe the prevalence, content, severity, interference and onset of OC symptoms in the postpartum.

## 2. Methods

### 2.1. Study Design

This was an observational (descriptive and psychometric) study, part of an ongoing research project entitled “Screening, prevention and early intervention in perinatal psychological distress—effectiveness of a new program in primary healthcare” (FCT/ PTDC/DTP-PIC/2449/2014).

### 2.2. Ethical Review

The present study was approved by the Ethical Committees of the Faculty of Medicine, University of Coimbra and the Coimbra Hospital and University Center (Ref.: CE-036-2017).

### 2.3. Procedure and Participants

The majority of participants were recruited while they were pregnant at Bissaya Barreto Maternity, at the Coimbra Hospital and University Centre. The inclusion criteria were being above 18 years old, absence of a medical condition, and being between 12 and 28 weeks of gestation. Recruitment took place in 2018 and 2019.

The nature, objectives and procedures of the investigation were explained, data confidentiality was guaranteed, and all participants who agreed to participate gave their written informed consent.

After delivery, women who had given written consent were contacted by phone between the third and the sixth months postpartum to set up an appointment.

Participants were asked to fill in a booklet, including sociodemographic questions, the Portuguese preliminary version of the POCS [17], the Perinatal Anxiety Screening Scale (PASS) [30,31] and the Postpartum Depression Screening Scale [32,33].

Approximately one-third of the meetings were at the medical centers (when the mothers took their babies for vaccination) or at their homes, but the majority of the interviews was conducted by phone, and the self-report questionnaires were filled out online or sent by mail. All the interviews were performed by experienced clinical psychologists or psychiatrists. If there were questions left blank or answered in an illogical way, they tried to validate the data by calling the subject. All participants presenting missing/illogical data were reachable.

All participants were interviewed with the Diagnostic Interview for Psychological Distress—Postpartum [33], a brief semi-structured diagnostic interview developed based upon the DSM-5 [16] criteria for the most prevalent perinatal psychiatric disorders, including the Obsessive–Compulsive Disorder.

The sample was composed of 212 women in postpartum, with a mean age of 33.13 ± 6.65 (range: 20–44) years; the mean age of the babies was 19.58 ± 9.97 (range: 12–40) weeks.

Participants were all native speakers of the Portuguese language. The majority were Portuguese (*n* = 137; 90.7%), married/living with a partner (*n* = 192; 90.5%), had High Scholl or College degree level (respectively, *n* = 67, 31.9%; *n* = 77, 36.4%) and were expecting the first (*n* = 112; 53.0%) or the second (*n* = 89; 42.4%) child.

### 2.4. Instruments

#### 2.4.1. Perinatal Obsessive–Compulsive Scale

We obtained the authorization to translate and validate the POCS from the author of the original form, Catherine Lord, who advised us to use the 2015 enhanced version [Unpublished manuscript]. The scale was translated into (European) Portuguese using a translation/back-translation method reported elsewhere [29]. Four questions related to the onset, course and duration of symptoms were added.

Women are asked to mark the presence or absence, as well as the time of onset of specific undesirable or troubling thoughts and behaviors. The POCS includes 19 thoughts (Section A) and 14 behaviors (Section B). To assess symptom severity, questions similar to those in the Y-BOCS are used to rate the amount of time spent, interference, distress, resistance and control of the disclosed thoughts and behaviors (severity scale: ten questions with scores ranging from 0 to 4, with higher scores indicating greater symptom severity). Information is also collected about how much the reported symptoms interfere with different aspects of participants’ lives, namely family, significant other, older child(ren), social activities, home responsibilities/housework, work responsibilities (interference scale: six questions with scores ranging from 0 to 4, total scores ranging from 0 to 28, with higher scores indicating greater symptom severity) (Section C). To score the POCS, only the severity questions from both the thoughts and behaviors are used. According to Lord [17], the score only serves as an indicator to seek more targeted help and is in no way a mean for establishing a diagnosis, just like the YBOCS. The sum of the thoughts and the behaviors severity scales (Sections A and B) can range from zero to 40, like the YBOCS.

#### 2.4.2. Postpartum Depression Screening Scale—Short Version

The majority of PDSS items correspond to cognitive-affective symptoms, which are more useful when assessing the presence of perinatal depression [34,35]. For each item, the woman is asked to rate the feelings that she has experienced during the last 2 weeks on a Likert scale from 1 (strongly disagree) to 5 (strongly agree). The PDSS short version (PDSS-21) presents good reliability and screening ability [31,32]. In the present study, the Cronbach alpha was 0.91.

#### 2.4.3. Perinatal Anxiety Screening Scale

The Perinatal Anxiety Screening Scale (PASS) [30] is composed of 31 items, with higher scores being indicative of more severe anxiety. Its items are also related to the perinatal context and were developed to systematically encompass international ICD and DSM diagnostic criteria for the various anxiety disorders. The Portuguese version presented good validity and reliability [31]. With the present sample, the Cronbach alpha was 0.95.

The PDSS-21 and the PASS were used as criteria for the POCS’ convergent validity because depressive, anxiety and OC symptoms commonly co-occur and may even convert into each other, particularly during the perinatal period [11,36].

#### 2.4.4. Diagnostic Interview for Psychological Distress-Postpartum (DIPD-PP)

DIPD-PP is a semi-structured clinical interview following a clinical approach of interviewing, where questions are grouped by diagnosis and criteria for a specific diagnosis. For the purpose of this study, we used the postpartum version, in which the temporal reference is “from birth”. DIPD-PP is considered a clinically relevant, practical and useful instrument in epidemiological and clinical research [37].

### 2.5. Statistical Analysis

SPSS 26.0 for Windows (Armonk: IBM SPSS), AMOS 26.0 (Chicago: IBM SPSS) and MedCalc Statistical Software version 19.2.6 (MedCalc Software bv, Ostend, Belgium; https://www.medcalc.org; 2020) were used. As the Shapiro–Wilk test proved that the majority of variables were not normally distributed, non-parametric measures and tests were applied. Descriptive statistics were used to describe demographics, prevalence and symptoms frequency and course.

AMOS 26.0 software was used to perform confirmatory factor analysis (CFA). The adjustment of the models was made from the modification indices higher than 11, *p* < 0.001. To evaluate the model fit, the following fit indices were used: Chi-square (χ^2^)/degrees of freedom (df), Comparative Fit Index (CFI), Goodness of Fit Index (GFI), Tucker–Lewis Index (TLI) and Root Mean Square Error of Approximation (RMSEA) [38]. Internal consistency was measured with Cronbach’s coefficient alpha. High values indicate consistent and reliable measures (0.65–0.70, acceptable; 0.70–0.80, good; 0.80–0.90, very good) [39].

Spearman correlations, Mann–Whitney U and Qui-Squared tests were applied to explore convergent and discriminant validity. The magnitude of the correlations was interpreted following Cohen’s criteria (1992): 0.01, small; 0.30, medium and 0.50, high. MedCalc was used to perform the Receiver Operative Characteristics (ROC) analyses in order to determine the cut-off score (adjusted to prevalence) with the best Youden Index and other parameters as the Area Under the Curve (AUC) and the conditional probabilities.

## 3. Results

### 3.1. Psychometric Study of the Postpartum Obsessive-Compulsive Scale

#### Construct Validity

Using Confirmatory Factor Analysis, we started by testing the final model obtained for the PreOCS [29], a second-order model, joining the Severity Sections A (Obsessions) and B (Compulsions) (10 items) into a single second-order factor. After correlating six pairs of errors, corresponding to items C1 and C2, C3 and C7, C3 and C5, C2 and C3, B2 and B4 and B5 and B6 all with modification indices higher than 12.00] (Figure 1), it presented an acceptable fit (χ^2^/df = 2.2971; CFI = 0.9319; GFI = 0.8574; TLI = 0.9127; RMSEA = 0.860, *p* < 0.001).

Spearman correlations between factors are presented in Table 1.

### 3.2. Reliability (Internal Consistency)

The POCS Cronbach’s alpha for the Severity Scale (ten items) was 0.91. For both, correlation coefficients between each item and the total score (excluding the item) were all high, with item A6 (*How much control do you have over the behavior*) being the lowest, *r* = 0.59.

For the Thoughts severity and the Behaviors severity scales, the Cronbach’s alphas were 0.82 and 0.92, respectively. All the items presented good internal consistency, with correlation coefficients with the corrected scores higher than 0.60.

For the POCS Interference scale (Section C, six items), Cronbach’s alpha was 0.90. Again, correlation coefficients between each item and the total score (excluding the item) were all high, being the lower coefficient that of item C3 (*relationships with your older child(ren)*), r = 0.56.

For all the scores, all the items contributed to their respective dimension internal consistency; that is, all had the effect of lowering Cronbach’s alpha if removed.

### 3.3. Convergent Validity

Spearman correlations of the POCS total score and Severity and Interference scales scores with the depression and anxiety scores are presented in Table 2. The correlations between the factors and between the factors and the total severity score were all significant, positive and of medium to high magnitude.

### 3.4. Criterion (Concurrent) Validity

Total and dimensional POCS scores, as well as symptom proportions (obsessions and compulsions), were compared by diagnostic groups. Women who met the diagnostic criteria for more than one disorder (major depressive disorder, anxiety disorder and/or OCD; *n* = 6; 2.8%) were excluded from this analysis.

The criteria were operationalized using DIPD-PPT/DSM-5 diagnoses, to compare four groups: Group 1—With only OCD, *n* = 4 (1.9%); Group 2—With only Major depression, *n* = 9 (4.2%); Group 3—With only anxiety disorder, *n* = 7 (3.3%); Group 4—Unaffected/without any of these disorders, *n* = 186 (87.7%).

As Table 3 summarizes, women with only OCD presented significantly higher POCS Severity mean scores than women with only major depression, with only anxiety disorder and unaffected. Regarding POC Interference, women with OCD presented significantly higher mean scores than unaffected women but did not differ from women with depression or with anxiety.

Concerning the proportion of symptomatic answers (obsessions and compulsions), Chi-squared tests were used. Groups 1, 2 and 3 were each compared with Group 4.

Repeated thoughts or pictures (Section A items) and behaviors (Section B items) in which Group 1 significantly differed from Group 4 are presented in Table 4. For all the obsessions, the proportion was significantly higher in women with OCD.

When comparing Group 2—With only Major depression and Group 4—Unaffected, any significant difference was found regarding the presence of obsessions and compulsions.

The comparison of Group 3—With only Anxiety disorder and Group 4 revealed that only items four items significantly differ: 4 *Somebody taking your baby away* [*n* = 65 (32.7%) vs. 5 (71.4%); X^2^ = 4.530; *p* = 0.044], 7 *Baby being harmed or dying in an accident* [*n* = 67 (33.7%) vs. 5 (71.4%); X^2^ = 4.241; *p* = 0.042], 8 *Baby acquiring a head injury* [*n* = 50 (25.1%) vs. 5 (71.4%); X^2^ = 7.408; *p* = 0.016], 11 *Screaming at your baby* [*n* = 22 (10.7%) vs. 3 (1.5%); X^2^ = 6.414; *p* = 0.040] and 12 *Harming your baby during bath time* [*n* = 23 (11.2%) vs. 3 (1.5%); X^2^ = 6.007; *p* = 0.044] from the Section A presented significantly higher proportions in affected women.

### 3.5. POCS Severity Scale Screening Performance

ROC analyses were applied to obtain the Area Under the Curve (AUC) to select the cut-off score (adjusted to prevalence) that potentiates the Youden Index and to determine the associated conditional probabilities in screening for P-POC. The POCS cut-off point (sum of the ten items evaluating severity) that maximized the Youden Index (J = 0.86, 95% CI [0.94–0.99]) was 20, which resulted in sensitivity of 85.71% (95% CI: 42.1–99.6%), specificity of 100.0% (95% CI: 98.2–100%), positive predictive value of 100 and negative predictive value of 99.5% (95% CI: 97.1–99.9%). The negative likelihood ratio was 0.14 (95% CI: 0.002–0.09). The AUC was 0.970 (*p* < 0.001; Standard Error = 0.031; 95% CI: 0.937 to 0.988, z = 15.360) (Figure 2). According to POCS cut-off, 20,205 women (96.7%) were non-cases, and 7 (3.3%) were cases of ppOCD.

### 3.6. Epidemiological Study

#### 3.6.1. OCD Diagnosis Prevalence

The prevalence of postpartum OCD was 3.30% (*n* = 7), major depression was 7.1% (*n* = 15) and anxiety disorder was 5.2% (*n* = 11).

#### 3.6.2. Symptoms Checklists

The percentage of postpartum women with OC symptoms was 75.9% (*n* = 161; only those filled in the Severity (Sections A and/or B) and Interference Scale (Section C)); with at least one repeated thought/picture (obsession) was 74.1% (*n* = 157) and with at least one repeated behavior (compulsion) was 41.5% (*n* = 88). Table 5 and Table 6 show the prevalence of obsessions and compulsions found in our sample.

### 3.7. Symptoms Severity and Interference Scales

Regarding the severity of thoughts, 10.2% (*n* = 16) of women were bothered by thoughts at least 1–3 h/day, and 3.2% (*n* = 5) more than 8 h/day. Concerning the interference with life and everyday activities, for 13.4% (*n* = 21), it was moderate, and for 3.2% (*n* = 5), it was severe (affecting the ability to function). Thoughts were moderately or severely distressing for 16.0% (*n* = 25) and extremely (disabling) distressing for 0.6% (*n* = 1). Finally, with regard to difficulty dismissing the thoughts, this was moderate/severe for 12.1% (*n* = 19), and 2% (*n* = 3) had little or no control over the thoughts.

Concerning compulsive behaviors, 28.4% (*n* = 25) were bothered by them more than 1–3 h/day, of which 6.6% (*n* = 6) were affected more than 3 h/day; for 22.7% (*n* = 20), behaviors interfere with life and everyday activities and were moderately or severe distressing for 12.5% (*n* = 11); also for 12.5% the difficulty to dismiss the thoughts were moderate or severe; 14.7% (*n* = 13) had little or no control over the thoughts.

### 3.8. Onset of Symptoms

Thirty-five percent of women (*n* = 55) date the onset of symptoms when questioned in the first three months postpartum, of which 13.4% (*n* = 21) around the time of birth of this child. For 12.1% (*n* = 21) of women, symptoms began during pregnancy, distributed evenly over the trimesters: one third (*n* = 7) when found out being pregnant with this child, one third (*n* = 7) during the first and second trimesters and the other third (*n* = 7) in the third trimester. For two participants (1.3%), symptoms started as soon as while planning pregnancy. Seven percent (*n* = 12) of women had already experienced OC symptoms in a previous perinatal period, 5.1% (*n* = 8) during a previous pregnancy, and 1.9% (*n* = 3) during the year postpartum of a first child.

Finally, 8.9% of women (*n* = 14) reported a specific trigger, such as after episodes of baby choking and/or breathing difficulty and hearing, watching or reading about families who lose newborns or babies due to sudden death or abduction.

Ten women (6.4%) already had excessive worries or repeated unpleasant thoughts/images (in general, not necessarily about pregnancy or the baby) prior to this pregnancy, of which fifty percent (*n* = 5) reported an increase in interference, severity and frequency; the other half did not notice any changes. Sixteen women (10.2%) maintained excessive worries or repeated thoughts/images at the time of the evaluation.

Considering the thoughts or behavior interference (Section C), 5.7% (*n* = 12) of women rated that they interfere “a lot” or “all the time” with the relationships with family; 9.5% (*n* = 20), 5.1% (*n* = 11), 5.7% (*n* = 11), 7.5% (*n* = 16), 16.6% (*n* = 35) and 13.7% (*n* = 29) registered this level of interference respectively with relationship with your significant other, relationships with your older child(ren) (when applicable), with the newborn baby, social activities, home responsibilities/housework and work responsibilities (when applicable).

## 4. Discussion

This work has been implemented to have a valid and reliable instrument allowing the assessment and screening of the OC phenomenology in postpartum, contributing to better detection and understanding of it.

We started to examine the POCS’ construct validity using CFA, which confirmed the adequacy of the measurement model. The acceptable fit obtained for the second-order model, joining the Severity Sections A (Obsessions) and B (Compulsions) (10 items), shows that these scales can be used as independent scores, for example, for research purposes, and may also, as Lord proposed, be put together to rate the general severity, which is particularly useful for screening proposes. The model also confirms that interference, which includes obsessions and compulsions, can be used as a general measure of ppOCD impact in the relevant life spheres.

The construct validity is also supported by the high intercorrelation between first order and second order factors. The internal consistency analysis, resulting in Cronbach’s alphas higher than 0.80, reinforced the homogeneity of all the items and the reliability of the scores.

The results concerning the convergent validity were as expected. Depression and anxiety symptomatology were used as criteria because their comorbidity with OC symptoms is well established [11,14,40,41,42]. It is known that mothers’ depression severity is positively related to various aspects of intrusive thoughts’ severity and interference [10,18,43].

When comparing the Behaviors severity scale with the Thoughts severity scale, we noticed that while the first (compulsive behaviors) correlated with a slightly higher magnitude of depression (measured by PDSS), the last (obsessive thoughts) showed higher correlation scores with anxiety (measured by PASS and its dimensions). This result indicates that, while compulsions may contribute to a transient relief of anxiety, they may also interfere with daily life leading to feelings of uncontrollability, higher ppOCD severity and depressive symptoms [44].

The group of OCD and related disorders has recently gained its own statute in psychiatric classifications instead of being included as anxiety disorders in DSM-5. This was done to highlight the differences between the different types of OCD disorders that a subject might present with but is not meant to ignore the prominence of the effect of anxiety that often appears with OCD symptoms. In line with this view, perinatal OCD is seen as a latter onset (the middle twenties) subtype with higher prevalence in the female gender and a close relationship with affective disorders [36]. This opposes child-onset OCD, which is more prevalent in male children and occurs with comorbid externalizing disorders (e.g., ADHD, tics). Our convergent validity analysis reinforces this view of ppOCD as presenting a specific clinical picture, which is intrinsically related to anxiety and depression.

Criterion (concurrent) validity was also analyzed through the comparison of POCS scores and symptoms proportions between women with and without ppOCD according to the DIPD/DSM-5 diagnosis. The much higher levels (more than double) of the OCD group’s severity and interference reinforce the validity of the instrument.

Additionally, the comparison of POCS scores with other diagnostic groups, such as women with depression and anxiety, favored the validity, as the POCS Severity was significantly higher in women with ppOCD than in women with postpartum major depression and postpartum anxiety disorders. These two groups (postpartum major depression and anxiety disorders) did not distinguish from each other in POCS severity nor from women not affected by these psychiatric disorders. The reduced size of the groups can be seen as a limitation of this study. However, it reflects the prevalence of the disorders in women in the perinatal period. The rigor of excluding cases with comorbidity (POC/major depression/anxiety disorders) further reduced the group size, but it was important to appreciate if POCS scores differentiate women with OCD from women without OCD and also from women with depressive and anxiety disorders, in which negative thoughts and images also occur.

In terms of POCS Interference, only women with ppOCD and unaffected women had significantly different scores. The low perceived interference of covert rituals, which are frequent during the perinatal period, may account for the lack of differences in OC symptoms of interference between newly mothers with OCD and affective disorders (anxiety and depression). Thus, higher severity of OC thoughts and behaviors, but not necessarily interference with daily routines, maybe a specific feature of postpartum OCD, at least in the early phases of the disease. On the other hand, this result exposes one potential limitation of the POCS, which is the lack of items evaluating avoidance behaviors, which are also prominent in postpartum.

The severity items of thoughts and behaviors evaluate parameters (e.g., time spent, control, distress) that are strongly related to the underlying processes, such as the habitual persistence of obsessions and compulsions following the OCD cycle. In this study, we intended to investigate if the content of thoughts and behaviors was also a prominent feature of mothers with OCD. When comparing the presence of specific symptoms by diagnostic groups, women with OCD presented significantly higher proportions in 7 (out of 22) obsessions and 6 (out of 12) compulsions than unaffected women, while the latter and women with major depression did not differ in this parameter, either in obsessions or in compulsions. Women with an anxiety disorder did not significantly differ from unaffected women in the proportion of any compulsion and only differed in the proportions of five obsessions, all of them related to the baby’s safety and health.

Considering the repeated thoughts that effectively discriminated between women with ppOCD versus unaffected women, the items *Harming your baby during bath time* (*p* = 0.008), *Burning the baby* (*p* = 0.009), *Harming your baby while he/she is asleep* (*p* = 0.001), *baby bleeding* (*p* = 0.036), *baby being spiritually possessed (for example by a negative force)* (*p* = 0.001) also presented overall lower prevalence in the total sample. While we may speculate that, when present, these obsessions may indicate a higher risk of developing or having ppOCD, instead of just being normative perinatal worries, this does not mean that these cognitions necessarily require medical intervention. Other investigators have also found that aggressive obsessions are the most prevalent in ppOCD [45,46,47,48,49]. Yet, none have reported subjects acting on these obsessions [13,36]. Instead, women with OCD reported reacting to these obsessions by performing washing and checking compulsions. The comparison between affective disorders (anxiety and depression) and unaffected women shows that only some repetitive thoughts and none of the repetitive behaviors distinguish healthy and unhealthy women. This indicates that the emergence of compulsions is specific to OCD, including postpartum OCD. While it is well established that cognitive dimensions are central in perinatal OCD, the presence of compulsive behaviors defines the disorder.

To develop an etiological model and treatment protocol for ppOCD, it is necessary to better characterize obsessions and compulsions in this context [36], namely their content, which differs from other OCD subtypes. Despite the differences between our sample and the sample of the original study, which was smaller and largely clinical, the most prevalent intrusive thoughts were the same (*nutrition for myself or my baby; dropping baby*; *baby dying in sleep*; *baby being harmed or dying in an accident*; *somebody taking the baby away*; *being criticized and/or judged as a mother*).

These contents relate to specific priorities and preoccupations of mothers during the postpartum period. Thus, the view of postpartum OCD as a dysfunctional process seems limited in that the specific content of thoughts must contribute to the associated distress leading to those repetitive thoughts’ persistence and conversion into true obsessions.

Another perspective of viewing the comparison of symptom proportions is that in 15 (out of 22) obsessions, women with and without ppOCD did not significantly differ. This and the similarity in the content of the most prevalent repeated intrusive thoughts of new mothers between clinical (original POCS study) and non-clinical (present study) samples suggest the existence of a continuum between normative and clinical perinatal experiences and reinforce the idea that repetitive intrusive thoughts may be nearly universal during the perinatal period, as proposed by cognitive behavioral models [4]. Therefore, clinical assessment (e.g., screening, intervention) in ppOCD should be supported in other parameters rather than the presence of thoughts, namely, as mentioned before, by their content, but also by severity and interference criteria.

The most prevalent repeated behaviors that also overlapped with the original study were checking the baby while she/he is asleep (25.9%); washing or cleaning your hands (18.9%); excessive searching (internet, books) about pregnancy, childbirth and babies (15.6%); and checking that you did not make a mistake (14.6%). Thus, in general, women endorsed less on repeated behaviors (compulsions) than repeated thoughts (obsessions), which may be explained by the fact that new mothers favor avoidance and covert rituals as control strategies for distressing thoughts [49]. We intend to address this POCS limitation in future studies, adding new items to evaluate this construct. The proposal that ppOCD is mainly cognitive and less behavioral in its nature [42,47] is also worth considering and may also contribute to explaining this result.

Considering symptoms’ severity and interference, globally, when the repetitive behaviors were present, they took more time and generated more interference than obsessions. It is noteworthy that almost 40% of women had at least one obsession and compulsion, which thus represents a subclinical phenotype of higher risk, as these cases are already qualitatively, but not quantitatively, similar to the clinical phenotype.

ROC curves from the original study suggested a discriminator score of 9, above which the POCS severity scale rating would become clinically relevant, with a sensitivity of 62% and specificity of 92%. Nevertheless, the authors warned that this score must be interpreted with caution and needs further validation. Our ROC analysis indicated a cut-off point of 20, which resulted in higher and very favorable parameters.

Because of this excellent accuracy, the prevalence of the disease in our sample was exactly equal to the caseness proportion according to the questionnaire cut-off, 3.3%. This ppOCD prevalence is similar to figures reported in the most rigorous studies [2,8,9]. The recruitment of participants in general maternity ensures the representativeness of our sample. This is important because if a screening instrument is validated with a sample that has a non-representative prevalence, a methodological risk exists of getting a higher PPV and a smaller NPV that will not be representative of the real clinical practice situation. This methodology also enables a better distinction between normative repetitive thoughts without mental disorder and pathological obsessions.

Having an instrument with good screening accuracy is important not only for the purpose of routine clinical assessment, as Abramowitz and collaborators (2010) recommend that should be conducted for perinatal OC symptoms as it is for perinatal anxiety and depression symptoms, but also because the distinction between normal and pathological obsessions/compulsions is particularly difficult during this period because they are not exclusive of OCD and can occur in non-clinical samples [43]. In fact, in our sample, the percentage of women with at least one obsession was 74.1% and with at least one repeated behavior was 41.5%. These figures are similar to those presented by Abramowitz et al. (2006, 2007), who found that more than 80% of new parents from the community reported obsessive thoughts [45,50]. Furthermore, obsessions and compulsions related to the baby are significantly more frequent in women with postpartum depression [14,41] than in non-depressed women. Thus, POCS can also be useful in establishing the differential diagnosis between depression and/or anxiety syndromes with OC symptoms and OCD with depressive and/or anxiety features.

As suggestions for future studies, we emphasize the importance of developing cross-cultural and cross-validation studies to examine the underlying latent constructs, particularly if we consider that we had to correlate some pairs of errors to achieve a good fit of the POCS measurement model, which can be considered a limitation.

Future studies should also develop prospective designs to analyze the risk factors for OC symptoms and disorders, as screening programs should include not only the detection of current symptoms but also the evaluation of psychosocial risk factors. Again, which concerns ppOCD, the knowledge about this topic is still scarce. We are planning to do this for both genders as we are also validating the POCS for new fathers. Despite being much less investigated in fathers than in mothers, research shows the presence of subclinical obsessive-compulsive symptoms in fathers during the perinatal period, with the prevalence comparable to mothers [51].

We consider this new instrument is an important contribution not only to improving ppOCD detection but also to better understanding of its etiology. POCS can also help in delineating subtypes, which have been argued to aid theoretical development, identification of vulnerability factors, prediction of clinical course and response to treatment [52]. A recent study suggests that most women only share their suffering when they are questioned and encouraged to report their perinatal-specific symptoms [1].

## 5. Conclusions

The OCD prevalence in this sample of Portuguese women in the postpartum period (3.3%) is in line with the best published estimates, as well as the evidence that for approximately 15% of women in the postpartum, the obsessions and compulsions are severe enough to cause clinically significant distress and interference.

The Portuguese version of POCS has good validity, reliability and accuracy and may be considered ready for use in both clinic and research fields. POCS not only provides a total score that can be especially useful for selecting women in postpartum that may need further psychiatric assessment but also provides specific information regarding symptoms and individual patterns experienced by each woman, which provides an opportunity for open dialogue between patients and their health care providers, to help normalization, destigmatization and personalized intervention.

## Figures and Tables

**Figure 1 ijerph-19-10624-f001:**
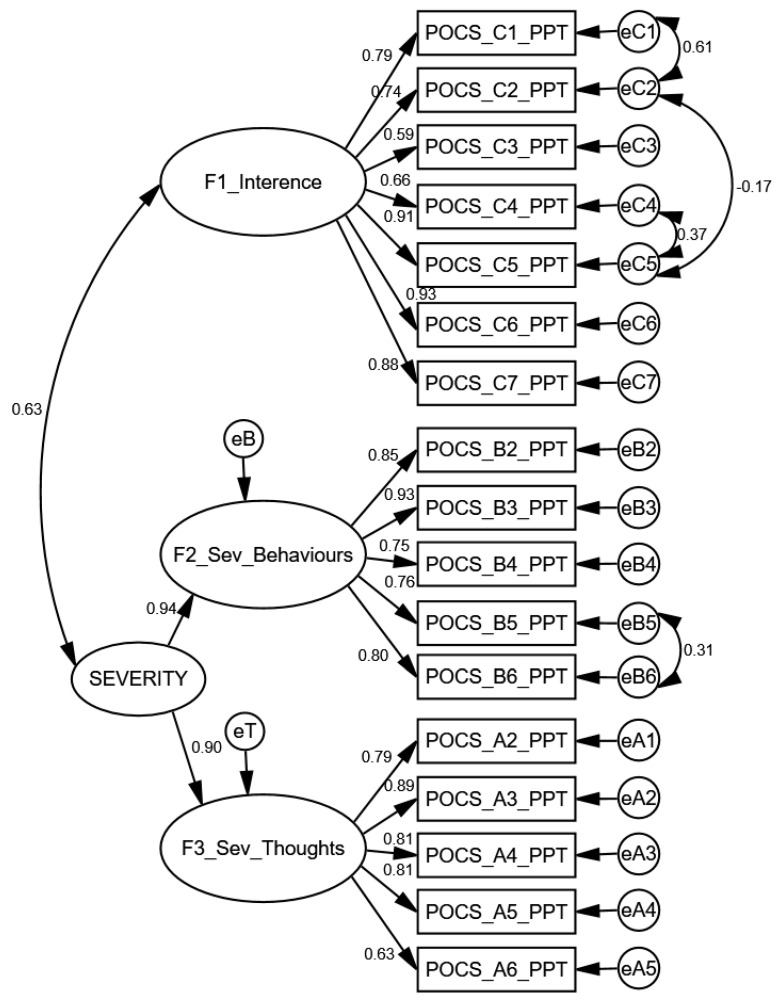
Final model of the POCS with standardized parameter estimates.

**Figure 2 ijerph-19-10624-f002:**
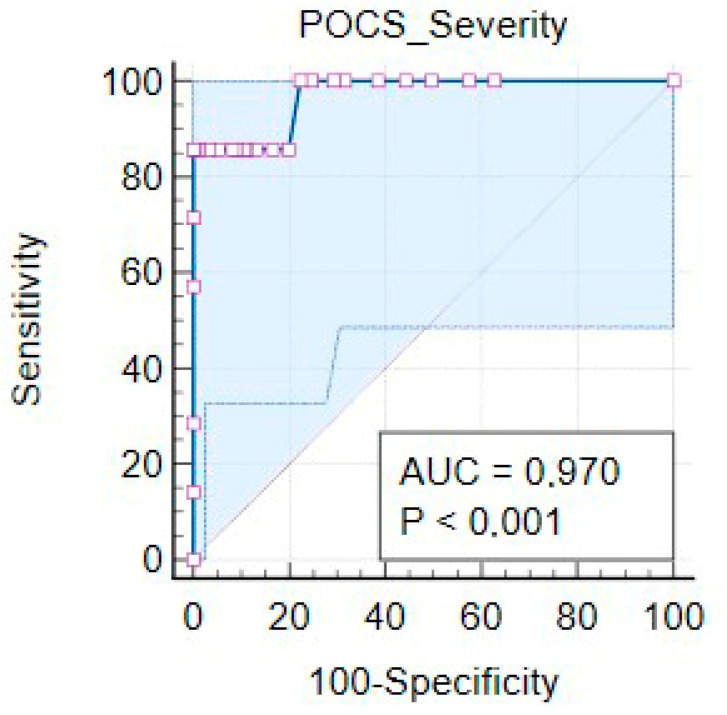
POCS ROC curve.

**Table 1 ijerph-19-10624-t001:** Spearman coefficient correlations between POCS factors.

	F1	F2	F3
F1 Interference			
F2 Behaviors severity	0.58; *p* < 0.001		
F3 Thoughts severity	0.50; *p* < 0.001	0.73; *p* < 0.001	
Total Severity Scale (F2 + F3)	0.57; *p* < 0.001	0.93; *p* < 0.001	0.93; *p* < 0.001

**Table 2 ijerph-19-10624-t002:** Spearman coefficient correlations of POCS with PDSS and PASS scores.

	PDSS (32.56 ± 12.41)	PASS (19.66 ± 15.59)
Thoughts Severity (4.17 ± 3.52)	0.38; *p* < 0.001	0.26; *p* < 0.001
Behaviors Severity (2.65 ± 233)	0.40; *p* < 0.001	0.24; *p* < 0.001
Total Severity (6.82 ± 6.31)	0.42; *p* < 0.001	0.27; *p* < 0.001
Interference (6.96 ± 6.90)	0.60; *p* < 0.001	0.38; *p* < 0.001

Note. Mean scores and standard deviations are given in parentheses. PDSS—Post-partum Depression Screening Scale; PASS—Perinatal Anxiety Screening Scale.

**Table 3 ijerph-19-10624-t003:** POCS scores by diagnostic groups/DSM-5—Mann–Whitney U.

POCS	Group 1 OCD *n* = 4 (1.9%)	Group 2 Depression *n* = 9 (4.2%)	Group 3 Anxiety *n* = 7 (3.3%)	Group 4 Unaffected *n* = 186 (87.7%)	Pairwise Comparisons (Z; *p*)
**Severity**					1 < 2 * (−2.476; 0.006)
M (SD)	18.75 (6.55)	5.11 (5.98)	6.86 (5.81)	4.41 (5.37)	1 < 3 * (−2.373; 0.006)
Md (IQR)	21 (10.71)	3 (8.50)	7 (7)	2 (7)	1 < 4 * (−3.162; 0.002)
**Interference**					1 < 4 * (−2.571; 0.003)
M (SD)	9.25 (5.50)	2.33 (4.79)	8.57 (10.61)	3.20 (5.88)	
Md (IQR)	12 (10)	0 (4)	4 (22)	0 (5)	

Note. M—mean; SD—standard deviation; Md—median; IQR—interquartile range. * *p* < 0.008 (Bonferroni correction).

**Table 4 ijerph-19-10624-t004:** Comparison of symptomatic answers between Group 1—With only OCD and 4—Unaffected/without any of these disorders—Chi-Squared Test.

	Group 1 OCD *n* = 4 (1.9%)	Group 4 Unaffected *n* = 186 (87.7%)	χ^2^; *p* OR (95% CI)
**Section A—Repeated worries, thoughts or images**			
4. Somebody taking your baby away	4 (100%)	66 (32.7%)	7.925; 0.013 1.061 (1.001–1.124)
9. Someone else having inappropriate sexual contact with your baby	3 (75.0%)	31 (15.3%)	10.128; 0.015 16.546 (1.667–164.280)
12. Harming your baby during bath time	3 (75.0%)	24 (11.9%)	5.168; 0.048 7.417 (0.998–55.119)
13. Burning the baby	2 (50.0%)	11 (5.4%)	13.169; 0.021 17.364 (2.231–135.152)
14. Harming your baby while he/she is asleep	2 (50.0%)	12 (5.9%)	12.021; 0.024 15.833 (2.049–122.372)
15. Your baby bleeding			
22. Baby being spiritually possessed (for example, by negative force)	2 (50.0%)	5 (2.5%)	26.988; 0.006 39.400 (4.582–338.775)
**Section B—Repeated behaviors**			
1. Washing or cleaning your hands	3 (75.0%)	34 (16.8%)	9.007; 0.003 14.824 (1.497–146.820)
3. Checking the door, locks, oven, etc.	4 (100%)	19 (9.4%)	32.456; <0.001 1.211 (1.004–1.460)
5. Checking that you did not make a mistake	3 (75.0%)	24 (11.9%)	13.721; 0.007 22.250 (2.224–222.582)
8. Washing and cleaning baby’s environment	2 (50.0%)	23 (11.4%)	5.485; 0.043 7.783 (1.045–57.937)
9. Bathing baby (ex. More than once a day)	1 (25%)	1 (0.5%)	24.498; 0.039 67.000 (3.345–341.813)
10. Checking the baby while she/he is asleep	4 (100%)	47 (23.3%)	12.398; 0.003 1.085 (1.002–2.176)
11. Combination of behaviors to prevent something bad from happening	2 (50.0%)	11 (5.4%)	13.169; 0.021 17.364 (2.231–135.152)

**Table 5 ijerph-19-10624-t005:** Prevalence of repeated thoughts or images in the postpartum considering total sample (*n* = 212).

Obsessions	*n* (%)
*Have you ever worried a lot or had repeated thoughts or pictures in your head about?*	
1. Being criticized and/or judged as a mother	70 (33.3%)
2. The nutrition for myself or my baby	100 (47.2%)
3. Baby being contaminated	59 (27.8%)
4. Somebody taking your baby away	74 (34.9%)
5. Dropping your baby	105 (49.5%)
6. Baby dying in her/his sleep	104 (49.1%)
7. Baby being harmed or dying in an accident	75 (35.4%)
8. Baby acquiring a head injury	58 (27.4%)
9. Someone else having inappropriate sexual contact with your baby	36 (17.0%)
*Have any of the following thoughts or images in your head repeatedly entered your mind, without implying that you would act on them?*	
10. Shaking your baby	21 (9.9%)
11. Screaming at your baby	25 (11.8%)
12. Harming your baby during bath time	29 (13.7%)
13. Burn the baby	15 (7.1%)
14. Harming your baby while he/she is asleep	16 (7.5%)
15. Your baby bleeding	16 (7.5%)
16. Throwing your baby	5 (2.4%)
17. Accidentally harming your baby with a sharp object/knife	8 (3.8%)
18. Stabbing your baby with a sharp object/knife	1 (.4%)
19. Choke the baby	7 (3.3%)
20. Squeeze the baby	6 (2.8%)
21. Inappropriate sexual contact with your baby	2 (.9%)
22. Baby being spiritually possessed (for example, by negative force)	8 (3.8%)
Other thoughts	4 (1.9%)

Note. The “other” thoughts included fear of baby catching diseases for which there is no vaccine yet and fear that the baby will choke and die for not being able to help.

**Table 6 ijerph-19-10624-t006:** Prevalence of repeated behaviors in the postpartum considering participants with at least one compulsion (*n* = 88).

Compulsions	*n* (%)
*Have you ever engaged in the following behaviors*	
1. Washing or cleaning your hands	40 (18.9%)
2. Strong urge to count or add	7 (3.3%)
3. Checking the door, locks, or oven, etc.	27 (12.7%)
4. Lining up and/or putting things in order	18 (8.5%)
5. Checking that you did not make a mistake	31 (14.6%)
6. Excessive searching (internet, books) about pregnancy, childbirth and babies	33 (15.6%)
7. Asking for reassurance	15 (7.1%)
8. Washing and cleaning baby’s environment	27 (12.7%)
9. Bathing baby (more than once a day)	3 (1.4%)
10. Checking the baby while she/he is asleep	55 (25.9%)
11. Combination of behaviors to prevent something bad from happening	14 (6.6%)
Other behavior	2 (.9%)

Note. The “other” behaviors included actions to ensure sleep hygiene and feeding routines.

## Data Availability

The data presented in this study are available on request from the corresponding author. The data are not publicly available due to privacy and ethics.

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
