# Peer review of "The Postpartum Obsessive-Compulsive Scale: Psychometric, Operative and Epidemiologic Study in a Portuguese Sample"

_ijerph, 2022, doi:10.3390/ijerph191710624_

Round 1

Reviewer 1 Report

The paper is good. Well-explained and interesting. However, I think that the results from discriminant validity should be interpreted with caution because the sizes of the comparison groups are too different and small. I don't think you can conclude that your scale have a good discriminant validity. Maybe you can delete this analysis or explain this, because it can lead to confusion.

Abstract:

In this section you should not use references or acronyms.

Line 27: DS should be written with the mean, not in bracket.

Line 42: you should write 'Conclusions' instead of 'Discussion'.

The abstract is too long.

Introduction:

Lines 55-58; 89-91 some words have a different font size.

Line 110: you should write the Parental Thoughts and Behaviours Checklist acronym so you can use it later on the text.

Line 114: the words 'was developed to' are duplicated. You should use the acronym of Perinatal Obsessive-Compulsive Scale, because you explained it in line 92.

Line 120: you should use the acronym for Yale- 120 Brown Obsessive-Compulsive Scale, because you explained it in line 100. Moreover, you use the name and year to cite the scale, and you should use a number.

Line 123: explain ROC acronym.

Methods: This section should be written in past tense. Please review.

Procedure. This section present information about participants' selection criteria that should be on Participants/Sample section. I think this section should be after instruments. Moreover, you should explain the design of your paper at the beginning of Methods section.

Participants. SD should be out of brackets. Line 177: school is misspelled.

Instruments. Line 199, reference is missed (Lord). Line 205, references numbers should be on the same bracket. Line 213 should be deleted.

Statistical analysis. Line 242, Spearman is misspelled.

Results: 

Line 249. If this is a subtitle, it should be listed. It is weird having a sentence  in between, not connected. 

Line 257. You write Fig 2 but the following figure is Figure 1.

Line 261. I think you should explain a little more about that analysis and the text should not be centered.

Reliability: items with a low correlation coefficient between each item and the total score should not be deleted? You should explain why you did not delete them.

Table 2: I think you should specify p scores for each correlation index instead of writing a table foot. The table should be reduced because there is so much blank and unnecessary space.

Discriminant validity: I'm not sure about these analysis because the groups are made up of a very different number of participants. OCD group only have 4 participants... You shoud explain this.

Tables 3 and 4. You should improve the presentation. The tables are too big for the collected information.

Lines 315-323. It is difficult to read these lines. Too many numbers. I think you should rewrite that paragraph.

Discussion

Lines 469-470. The font size is bigger.

You should include limitations.

Conclusions

You should not use references in this section. You should explain the main ideas extracted from your study. You should include some limitation and some future research.

Author Response

Thank you very much for the pertinent and useful revision and for the opportunity of improving our paper entitled “The Postpartum Obsessive-Compulsive Scale: Psychometric, operative and epidemiologic study in a Portuguese sample”.  We have appreciated the comments and have accepted the suggestions.

Please find below our answers and list of changes list of changes point by point.

The paper is good. Well-explained and interesting. However, I think that the results from discriminant validity should be interpreted with caution because the sizes of the comparison groups are too different and small. I don't think you can conclude that your scale have a good discriminant validity. Maybe you can delete this analysis or explain this, because it can lead to confusion.

After reflecting more deeply on the issues related to discriminant validity, we recognized that the term was not well applied. In the revised version, instead of discriminant validity we use 3.4. Criterion (concurrent) validity.

Criterion validity is an estimate of the extent to which a measure agrees with a gold standard (i.e., an external criterion of the phenomenon being measured), which in this case is the clinical diagnosis.  Criterion validity is concurrent when the scores of a test and criterion variables are obtained approximately at the same, what was the case (McIntosh et al. 2010).

MCINTOSH, A. M.; SHARPE, M.; LAWRIE, S. M. Companion to psychiatric studies, Research methods, statistics and evidence-based practice. 2010.

Abstract:

In this section you should not use references or acronyms.

References have been cut; the use of abbreviations was limited.

Line 42: you should write 'Conclusions' instead of 'Discussion'.

This was done.

The abstract is too long.

The size of the abstract was reduced to 300 words.

Introduction:

Lines 55-58; 89-91 some words have a different font size.

This has been fixed.

Line 110: you should write the Parental Thoughts and Behaviours Checklist acronym so you can use it later on the text.

This designation is not used again in the text.

Line 114: the words 'was developed to' are duplicated. You should use the acronym of Perinatal Obsessive-Compulsive Scale, because you explained it in line 92.

This has been fixed.

Line 120: you should use the acronym for Yale - Brown Obsessive-Compulsive Scale, because you explained it in line 100. Moreover, you use the name and year to cite the scale, and you should use a number.

This has been fixed.

Line 123: explain ROC acronym.

This was done - Receiver operating characteristic

Methods: This section should be written in past tense. Please review.

The entire section has been revised, to be written in the past, except the description of the scales used in the study (POCS, PDSS, PASS).

Procedure. This section present information about participants' selection criteria that should be on Participants/Sample section. I think this section should be after instruments. Moreover, you should explain the design of your paper at the beginning of Methods section.

We merged the Procedure and Participants sections into a single title - 2.1. Procedure and participants.

The study design is now mentioned at the beginning of Methods section - observational (descriptive and psychometric) study.

Participants. SD should be out of brackets. Line 177: school is misspelled.

This has been fixed.

Instruments. Line 199, reference is missed (Lord). Line 205, references numbers should be on the same bracket. Line 213 should be deleted.

This has been fixed.

Statistical analysis. Line 242, Spearman is misspelled.

This has been fixed.

Results: 

Line 249. If this is a subtitle, it should be listed. It is weird having a sentence in between, not connected. 

It was listed as a sub-title.

Line 257. You write Fig 2 but the following figure is Figure 1.

This has been fixed.

Line 261. I think you should explain a little more about that analysis and the text should not be centered.

This has been done.

Reliability: items with a low correlation coefficient between each item and the total score should not be deleted? You should explain why you did not delete them.

We apologize for not understanding this comment, as all correlations are high, contributing to the items’ internal validity. What we wrote was (lines 261 - 262): “correlation coefficients between each item and the total score (excluding the item) were all high, being the item A6 (How much control do you have over the behaviour) the lowest, r=0.59.”

Table 2: I think you should specify p scores for each correlation index instead of writing a table foot. The table should be reduced because there is so much blank and unnecessary space.

We have made the suggested changes.

Discriminant validity: I'm not sure about these analyses because the groups are made up of a very different number of participants. OCD group only have 4 participants... You should explain this.

As previously mentioned, the “Discriminant validity” term was replaced by “Criterion (concurrent) validity “.

The reduced size of the groups reflects the prevalence of the disorders in women in the perinatal period from the general population. The rigor of excluding cases with comorbidity (POC/Major depression/Anxiety disorders) further reduced the groups size. This analysis was important to show that POCS scores differentiate women with POC not only from women without POCS, but also from women with major depressive and anxiety disorders, in which negative thoughts and images also occur.

Nevertheless, these aspects were mentioned as potential limitations in the Discussion section (lines 439-443).

Tables 3 and 4. You should improve the presentation. The tables are too big for the collected information.

We made the suggested changes.

Lines 315-323. It is difficult to read these lines. Too many numbers. I think you should rewrite that paragraph.

The paragraph was rewritten. Numbers of the odds ratios were excluded to simplify.

Discussion

Lines 469-470. The font size is bigger.

This has been fixed.

You should include limitations.

Limitations were added (lines 509-511).

Conclusions

You should not use references in this section. You should explain the main ideas extracted from your study. You should include some limitation and some future research.

The only reference in this section has been removed, as well as the corresponding sentence.

Future research (lines 548-556) and limitations was already included and we have added other ideas.

Reviewer 2 Report

You put a lot of effort into conducting the research and writing the manuscript.  However, the impact factor of this journal is very high and high-quality manuscripts are being published, so it is necessary to follow the guidelines well. 

I have a question about a few things. It is necessary to organize the manuscript concisely.

1. It is necessary to indicate when the year in which the funding was received. year 2014?, 

2. I think the year I received the IRB was 2017. About 5 years pass between the data collection period and the derivation of the results, and the research results may become outdated.

3. A few things need to be corrected.

- abstract: It is necessary to follow the guidelines. <=300 words limit, now 450 words

- It is necessary to distinguish between the description in the introduction and the description in the discussion. Revision of introduction description is required.

- Separate the purpose of the study so that it is clearly visible.

- Please specify the research period. 2018 from - to -

- Separate each section for ethical review in the Methods section and present it in detail.

- Briefly describe the instrument with only the main contents.

- No mention of study design.

- Where can I find the results for DIPD-PP?

- References should be used as papers published within the last 5 years.

Author Response

Thank you very much for the pertinent and useful revision and for the opportunity of improving our paper entitled “The Postpartum Obsessive-Compulsive Scale: Psychometric, operative and epidemiologic study in a Portuguese sample”.  We have appreciated the comments and have accepted the suggestions.

Please find below our answers and list of changes list of changes point by point.

You put a lot of effort into conducting the research and writing the manuscript.  However, the impact factor of this journal is very high and high-quality manuscripts are being published, so it is necessary to follow the guidelines well. 

I have a question about a few things. It is necessary to organize the manuscript concisely.

  1. It is necessary to indicate when the year in which the funding was received. year 2014?

The reference for the application to obtain funding is from 2014. Due to successive delays, it only started in 2017.  This information has been added (line 583).

  1. I think the year I received the IRB was 2017. About 5 years pass between the data collection period and the derivation of the results, and the research results may become outdated.

Data collection took place in 2018 and 2019. This information has been corrected.

  1. A few things need to be corrected.

- abstract: It is necessary to follow the guidelines. <=300 words limit, now 450 words

The abstract was reduced to 300 words.

- It is necessary to distinguish between the description in the introduction and the description in the discussion. Revision of introduction description is required.

Introduction was revised.

- Separate the purpose of the study so that it is clearly visible.

Done.

- Please specify the research period. 2018 from - to –

It is now specified: “Recruitment took place in 2018 and 2019” (line 152).

- Separate each section for ethical review in the Methods section and present it in detail.

Done.

- Briefly describe the instrument with only the main contents.

The description of the instruments has been shortened.

- No mention of study design.

Study design is now mentioned: “This observational (descriptive and psychometric) study was part of…” (line 144)

- Where can I find the results for DIPD-PP?

Results for DIPD-PP can be found here:

Pereira, A.T.; Marques, C.; Xavier, S.; Azevedo, J.; Soares, M.J.; Bento, E.; Marques, M.; Nogueira, V.; Macedo, A. Prevalence and incidence of postpartum major depression (DSM-5) in Portuguese women. Postpartum Depress. Prevalence, risk factors outcomes 2017, 61–84.

- References should be used as papers published within the last 5 years.

References were updated, eight of which were replaced by studies published in 2021 and 2022, which also allowed reviewing the content of the introduction. In the revised version, only a few very specific references and those relating to the instruments are from more than five years ago.

Reviewer 3 Report

Please clarify how the authors obtained informed consent from participants of this Survey. Which test was used to test normality 'distribution? Discussion: Please discuss the limitations of the study. Literature: Newer literature should be used. Key words: MeSH indexed key words should be used

Prior to publication, more structuring and checking of language and grammar is suggested.

Author Response

Thank you very much for the pertinent and useful revision and for the opportunity of improving our paper entitled “The Postpartum Obsessive-Compulsive Scale: Psychometric, operative and epidemiologic study in a Portuguese sample”.  We have appreciated the comments and have accepted the suggestions.

Please find below our answers and list of changes list of changes point by point.

Please clarify how the authors obtained informed consent from participants of this Survey. Written informed consent was obtained. This information has been added.

Which test was used to test normality 'distribution? Shapiro-Wilk test was used to test normality. This information has been added.

Discussion: Please discuss the limitations of the study. Limitations are now more clearly discussed (lines 439-443; 509).

Literature: Newer literature should be used.

References have been updated; eight were replaced by studies published in 2021 and 2022; only a few very specific references and those relating to the instruments are from more than five years ago.

Key words: MeSH indexed key words should be used.

Done.

Prior to publication, more structuring and checking of language and grammar is suggested.

The language and grammar were reviewed by a bilingual colleague.

Reviewer 4 Report

Well written paper. No critiques and/or specific comments for this time.

Author Response

Thank you very much.

Round 2

Reviewer 2 Report

1. It is necessary to reduce the number of words in abstract to less than 300 words. It's still over 300 words. (337 words)

2. It is necessary to update the references to the latest papers. Of course, it may not be possible for some papers (instrument refereces et al.), but it is necessary to use papers within the last 10 years if possible. Some papers have been updated, but one more update is needed.

3. Do not describe the study design by including it in (Procedure and participants), but give it a lower number of methods ( ex. 2.1. study design).

4. Describe ethical review as a sub-number of methods. (ex. 2.4. ethical review)

Author Response

Thank you very much for the for the opportunity of continuing improving our paper entitled “The Postpartum Obsessive-Compulsive Scale: Psychometric, operative and epidemiologic study in a Portuguese sample”.  

Please find below our answers and list of changes:

  1. It is necessary to reduce the number of words in abstract to less than 300 words. It's still over 300 words. (337 words)

The number of words in the abstract has been reduced to 299.

  1. It is necessary to update the references to the latest papers. Of course, it may not be possible for some papers (instrument refereces et al.), but it is necessary to use papers within the last 10 years if possible. Some papers have been updated, but one more update is needed.

We have updated an additional twelve references in this second revision. In this version, the only references which are not within the last 10 years  are those relating to the instruments.

  1. Do not describe the study design by including it in (Procedure and participants), but give it a lower number of methods (ex. 2.1. study design).

This was done (lines 144-149).

  1. Describe ethical review as a sub-number of methods. (ex. 2.4. ethical review)

This was done (lines 150-153).

Best regards

This manuscript is a resubmission of an earlier submission. The following is a list of the peer review reports and author responses from that submission.